# Evaluation of a Yeast Hydrolysate from a Novel Strain of *Saccharomyces cerevisiae* for Mycotoxin Mitigation using In Vitro and In Vivo Models

**DOI:** 10.3390/toxins14010007

**Published:** 2021-12-22

**Authors:** Paul Gerard Bruinenberg, Mathieu Castex

**Affiliations:** 1Trouw Nutrition R&D, Stationsstraat 77, 3811 MH Amersfoort, The Netherlands; 2Lallemand SAS, 19 rue des Briquetiers, BP 59, CEDEX, 31702 Blagnac, France; mcastex@lallemand.com

**Keywords:** deoxynivalenol, mycotoxins, swine, toxicokinetic model, yeast hydrolysate

## Abstract

Mycotoxicoses in animals are caused by exposure to mycotoxin-contaminated feeds. Disease risk is managed using dietary adsorbing agents which reduce oral bioavailability. The objective of this work was to evaluate the efficacy of three selected yeast products as mycotoxin binders using in vitro and in vivo models. Their capacity to adsorb deoxynivalenol (DON), zearalenone (ZEA), and ochratoxin A (OTA) was evaluated using an in vitro model designed to simulate the pH conditions during gastric passage in a monogastric animal. Results showed that only one product, an enzymatic yeast hydrolysate (YHY) of a novel strain *Saccharomyces cerevisiae*, adsorbed about 45% of DON in solution. Next, we determined the effect of YHY on oral absorption of a DON, ZEA, and OTA mixture using a toxicokinetic model in swine. Toxicokinetic modeling of the plasma concentration-time profiles of DON, OTA, and zearalenone-glucuronide (ZEA-GlcA) showed that YHY tended to reduce the maximal plasma concentration of OTA by 17%. YHY did not reduce oral bioavailability of OTA, DON, and ZEA-GlcA. Within the context of this experiment, and despite some positive indications from both the in vitro and in vivo models employed, we conclude that the YHY prototype was not an effective agent for multiple mycotoxin adsorption.

## 1. Introduction

In the context of food and feed, mycotoxins are toxic secondary metabolites formed by fungi growing on agricultural commodities. Mycotoxin contamination is a serious threat to human and animal health and results in considerable economic losses to global animal husbandry. Mycotoxicoses are diseases caused by exposure to mycotoxin-contaminated feeds. In animal production systems, consumption of mycotoxin-contaminated feed can negatively affect performance through decreased feed efficiency, feed refusal, and reduced weight [1,2].

The *Fusarium* mycotoxin DON is the most frequently occurring mycotoxin in the temperate regions of North America and Western Europe. Other mycotoxins produced by *Fusarium* include ZEA and fumonisins (FB1 & 2). OTA produced by *Aspergillus* and *Penicillium* species is the main contaminant of cereals and soybeans. OTA prevalence and average concentration were reported highest in South Asia. Aflatoxins (AFB1 & B2), produced by *Aspergillus* species, are frequently found in cereal grains, such as maize, and are most prevalent in South Asia, South-East Asia, and Southern Europe [3,4]. Mycotoxin exposure induces a host of biological responses in animals, and, whole, some mycotoxins are classified as carcinogenic. DON and ZEA are of particular concern for swine due to their high susceptibility. DON consumption is characterized by immunosuppression and reduction of the barrier function of intestinal epithelium [5]. ZEA is best known for its toxic effect on fertility and reproduction in swine: it induces an estrogenic effect on the animal [6]. FB1 exposure has been implicated with lung pneumonitis and pulmonary oedema [7]. The primary target organ for OTA is the kidney; it is known to induce nephropathy in pigs and poultry [8].

Previous in vitro studies have shown that yeast cells can bind mycotoxins through physical adsorption, ion exchange, and complex binding structures on the cell surface and cell wall polysaccharides (β-D-glycan, glucomannan) have been identified as specific binding sites [9,10,11,12,13,14].

In feed, administration of mycotoxin binders results in adsorption of mycotoxins in the gastrointestinal tract, which should result in reduced absorption into the systemic circulation. However, there is little published data available in the literature on the effect of mycotoxin binders on the in vivo absorption of mycotoxins. In piglets administered an oral mycotoxin-contaminated bolus with or without a binder, the agricultural byproduct white grape pomace reduced significantly urinary mycotoxin biomarkers for AFB1 and ZEA [15]. The European Food Safety Authority (EFSA) stipulates that in vivo testing of mycotoxin binders is necessary to evaluate the efficacy of mycotoxins inactivators. However, in vivo studies where non-specific parameters such as body weight, feed intake, or feed conversion rate are measured exclusively do not meet the prerequisite criteria from EFSA to demonstrate efficacy of these products [16,17]. Toxicokinetic studies, based on absorption, distribution, metabolization, and excretion (ADME) of the mycotoxin, are necessary to evaluate the possible effects on the absorption of the toxin in pigs [18]. Similar approaches have notably been used to assess the impact of mycotoxin binders on the relative oral bioavailability of coccidiostats in pigs and chickens [19,20]. The goal of this work was to evaluate the in vitro efficiency of three yeast products as mycotoxin binders and to evaluate the efficacy of a selected hydrolysate on the oral absorption of DON, OTA, and ZEA using a toxicokinetic model in swine.

## 2. Results

### 2.1. In Vitro Adsorption of Mycotoxins by Yeast Products

In order to assess the mycotoxin binding capacity of three selected yeast products, the in vitro adsorption/desorption test was used as described in Sabater et al. [21]. This test was designed to mimic the temperature, pH, and passage time through the stomach and gut of a monogastric animal. Briefly, the mycotoxin adsorption capacity of adsorbents was tested by sequential incubations, each step for 2 h at 37 °C under constant agitation, starting at pH 3.0, and followed by pH 5.0 and 8.5. Table 1 and Appendix A show the results of in vitro DON, OTA, and ZEA adsorption by the yeast products at the pH levels tested. The yeast enzymatic hydrolysate YHY was able to adsorb about 45% of DON independent (*p* = 0.2001) of the pH value of the buffer during incubation steps. In contrast, the yeast autolysate products YA1 and YA2 displayed a very low DON adsorption (<12%), which was not significantly different from the control incubations (negative control, NC) without any adsorbent (*p* > 0.05). None of the yeast hydrolysates showed significant adsorption of OTA (*p* ˃ 0.1). Contrary to YHY, the products YA1 and YA2 both showed a statistically significant adsorption (*p* < 0.0001) in the range of 49.7% to 64.7% of ZEA, which was dependent (*p* < 0.0001) on pH value of the buffer. Effective adsorption of DON by the new yeast ingredient YHY could reduce the oral bioavailability of DON in farm animals, especially key for swine [5], and therefore YHY was selected for further evaluation of its efficacy. Studies concerning in vitro binding between mycotoxins and adsorbents are not always a reliable predictor of in vivo efficacy [15,22]. Therefore, to evaluate the in vivo efficacy of YHY, we have measured specific toxicokinetic parameters using the oral bolus model developed for swine [23].

### 2.2. Toxicokinetic Study

In conjunction with DON, the mycotoxins ZEA and OTA were also included in the in vivo toxicokinetic study, due to their relevance for mycotoxicoses in swine [6,8]. The effect of YHY on specific toxicokinetic parameters and relative oral bioavailability after a single oral bolus containing DON, ZEA, and OTA was determined. The control feed was shown to be contaminated with significant levels of DON, namely 1170 µg/kg feed, whereas the maximum guidance level is 900 µg/kg feed [18]. This indicates that all animals were exposed to the mycotoxin for one week prior to the bolus administration. The animals were deprived from feed 12 h prior to the bolus administration and this resulted in the absence of DON, as well as OTA and ZEN-GlcA, in plasma before bolus administration in all animals including the control group (see below).

No adverse effects were observed during the animal trial following bolus administration of the three mycotoxins. Figure 1a shows the plasma concentration-time profile of DON following oral administration of DON, whether or not combined with YHY (see Appendix A). Each profile represents the mean of six animals ± standard deviation (SD).

The toxicokinetic parameter responses to DON challenge are shown in Table 2 and Appendix A. The mean (± SD) AUC_0–8 h_ was 74.57 ± 8.39 h.ng/mL for DON and 60.08 ± 12.49 h.ng/mL for DON combined with YHY. With regard to the absorption phase of DON, a physiologically relevant parameter which occurs in the first 2 h following administration, the mean AUC_0–2 h_ was 30.03 ± 3.87 for 25.76 ± 4.58 for DON combined with YHY. No statistically significant differences for any of the toxicokinetic parameters were observed between groups. The relative oral bioavailability AUC_0–8 h_ and AUC_0–2 h_ for YHY was 80.50% and 85.79%, respectively. The maximum plasma concentration C_max_ for DON was numerically reduced by 23% by adsorbent YHY (*p* = 0.306).

Figure 1b shows the plasma concentration-time profile of OTA after oral bolus administration (Appendix A). Table 3 shows the derived toxicokinetic results of OTA following oral bolus administration, with or without YHY (Appendix A). The mean (±SD) AUC_0–96 h_ of OTA was 14.13 ± 1.48 h.µg/mL for OTA and 12.28 ± 2.06 h.µg/mL for OTA combined with YHY. The timespan of absorption of OTA was longer in comparison with that of DON, as was demonstrated by the T_max_ value (2.42 and 1.13 h, respectively). In the first 4 h following administration, the mean (±SD) AUC_0–4 h_ was 1.42 ± 0.24 h.µg/mL for OTA and 1.14 ± 0.25 h.µg/mL for OTA combined with YHY. Moreover, clearance of OTA from the blood was much slower in comparison with DON and ZEA-GlcA, because OTA is reported to bind plasma proteins [24]. This resulted in the observed elimination half-life T_1/2 el_ for OTA of 45.12 h ± 6.61, whereas for DON and ZEA-GlcA, T_1/2 el_ was 2.15 h ± 0.10 (Table 2) and 3.60 h ± 2.00 (Table 4), respectively. In the case of OTA, the relative oral bioavailability AUC_0–96 h_ and AUC_0–4 h_ for YHY was 86.86% and 79.86%, respectively. No statistically significant differences for the toxicokinetic parameters of OTA were observed between the control group and the group receiving YHY, with the exception of a tendency (*p* = 0.076) for maximum plasma concentration C_max_ that was reduced by 17% with the addition of YHY (Table 3).

Following ZEA administration, all plasma concentrations were below the limit of quantification of the method of 0.5 ng/mL. Therefore, ZEA-GlcA, the major ZEA phase II metabolite, was selected as biomarker in blood plasma for ZEA exposure. To enable the quantification of ZEA-GlcA, ZEA was administered at a relatively high dose of 0.5 mg ZEA/kg BW. In Figure 1c, the chromatographic peak area-time curve is presented of ZEA-GlcA after oral bolus administration of ZEA (Appendix A).

Table 4 shows the results of the most relevant toxicokinetic parameters of ZEA-GlcA after oral bolus administration of ZEA, with and without YHY inclusion (Appendix A). The mean (±SD) AUC_0–8 h_ of ZEA-GlcA was 2168.20 ± 494.87 h.peak area/mL for ZEA and 1917.16 ± 444.98 h.peak area/mL for ZEA combined with YHY. ZEA demonstrated a short absorption phase of 0.5 h, and the AUC_0–0_._5 h_ of ZEA-GlcA was 505.92 ± 227.89 h.peak area/mL for ZEA and 407.48 ± 141.70 h.peak area/mL for ZEA combined with YHY. The relative oral bioavailability AUC_0–8 h_ and AUC_0–0_._5 h_ for YHY was 88.42 and 80.54, respectively. No statistically significant differences or tendencies for any of the toxicokinetic parameters were observed between the treatment groups. Addition of YHY to the oral bolus numerically reduced the maximum plasma concentration C_max_ of ZEA-GlcA by 20% (*p* = 0.308).

## 3. Discussion

The present work examined the efficacy of the selected yeast product YHY to adsorb the predominant mycotoxins in vitro and in vivo. This bioactive YHY comprises dehydrated enzymatically hydrolyzed yeast cells of a selected strain of *S. cerevisiae*, which includes the cell wall fraction. The yeast cell wall represents about 30% (*w*/*w*) of the weight of the cell. The inner layer is composed of the polysaccharides β-1,3-glucan and highly branched β-1,6-glucan, which together with chitin represents 50–60% of cell wall dry weight and is providing mechanical strength to the wall [11]. The outer cell wall layer is composed of heavily glycosylated mannoproteins, which are involved in cell–cell recognition events and limitation of wall porosity [25,26]. Several studies demonstrated that the yeast cell wall components exhibit many different adsorption sites, as well as different binding mechanisms with mycotoxins such as hydrogen bonds, ionic or hydrophobic interactions, and van der Waals interactions [9,10,11,12,27]. Yeasts and yeast cell wall extracts have shown in vitro adsorption efficacy for a number of mycotoxins, including ZEA, AFB1, T2-toxin, patulin, and OTA [8,11,12,13,28]. However, there are only limited and conflicting reports on DON adsorption by yeasts or by products derived therefrom. It has been reported that DON had an in vitro affinity of around 30% with pure model β-D-glucans at pH 3.0 and 6.0, but no affinity at pH 8.0 [12]. Furthermore, in vitro studies have shown that a commercial preparation of *S. cerevisiae* mannans has the capacity to adsorb DON [22]. In the latter study, the amount of DON adsorbed in buffers at pH 3 and 7 was between 80 and 90% with a DON and adsorbent concentration of 1–2 µM (0.3–0.6 ppm) and 1 mg/mL, respectively. Additionally, three commercial organic adsorbents (glucomannan, modified yeast cell wall, and esterified glucomannan), tested at 2 mg/mL in an in vitro model with buffer solutions at pH 3 and pH 7, showed 36–56% adsorption of DON (assayed at 0.9 ppm) [29]. In comparison, our study showed a significant (*p* < 0.0001) in vitro DON adsorption of about 45% carried out in buffers at pH 3, 5, and 8.5 with a concentration of 1 ppm for DON and 2 mg/mL for the adsorbent YHY. The kinetic profile of various trichothecenes, including DON, was investigated during a four-day fermentation with the lager yeast *S. pastorianus* [30]. The authors reported that during the first 4 h of fermentation, this yeast removed about 13% of the DON from the brewer’s wort. DON removal was reported to be likely due to binding of DON to yeast cells, since after a steady decline over the first 24 h, the DON concentration stabilized at ca. 85% for the remainder of the fermentation period. The manufacturing process of YHY involved a heating and roller drying step, effectively inactivating the processing enzyme and endogenous yeast enzymes, and, hence, suggesting a binding mechanism for DON removal. In the present in vitro study, the stability of the DON-YHY complex was numerically unaffected (*p* = 0.2001) at a pH scale ranging from 3.0 to 8.5, suggesting the absence of cation exchange mechanism and a more major role of hydrophobic interaction in the binding [14].

The capability of YHY to adsorb DON was further examined using a toxicokinetic model, which indicated that YHY, at a realistic inclusion level corresponding to 2 mg/kg feed, had a limited effect on the absorption of DON, as the maximal plasma concentration and oral bioavailability of DON was only numerically reduced by about 20%. This discrepancy confirms earlier observations of other research groups that in vitro binding between adsorbent and mycotoxins seems to have a limited utility in predicting the in vivo efficacy of binders [15,22].

DON is mainly absorbed in the proximal part of the small intestine by means of passive diffusion [31]. Nevertheless, the present toxicokinetic study showed that a yeast hydrolysate comprising cell wall components was able to reduce the DON plasma concentration in piglets, which, according to the EFSA [18], is the most relevant parameter to evaluate the efficacy of a mycotoxin binder. Additionally, the toxicokinetic study showed that YHY reduced the maximal plasma concentration C_max_ for OTA and ZEA-GlcA by 17% and 20%, respectively.

It is important to note that the pig toxicokinetic model system used in this study has limitations. In our study, the SD values for important parameters such as maximal plasma concentration and oral bioavailability for DON, OTA, and ZEA-GlcA were ranging between 10 to 45%, suggesting that the high variability in these measurements limited our ability to determine statistically meaningful differences. Previous studies using the same mycotoxin oral bolus toxicokinetic pig model reported comparable variability for maximal plasma concentration and oral bioavailability values of DON [23,32,33,34].

An additional limitation in the current model was the administration of the DON-contaminated feed one week prior to evaluation of the YHY. During this time, we cannot rule out that the animals mounted a response, e.g., by alteration of the barrier function of intestinal epithelium [5], that would have masked the benefits induced with the introduction of the YHY. It is reported that pigs have the ability to adapt to the presence of DON in the diet [35]. Another study showed that a primary decreased weight gain in pigs fed a DON contaminated diet (1 mg/kg), which is a similar DON contamination level as in our study, in the first week was compensated thereafter [36]. The adaptive mechanism of pigs to DON has not been fully understood. This resistance may be attributed to the effect of DON in the gut and especially its ability to alter the composition of intestinal microbiota [35].

Furthermore, the effect of different dosages of mycotoxin and binder in the oral bolus model was not explored. Future work using this experimental approach merits increased sample size to account for the inherent variability in toxicokinetic evaluations.

It cannot be excluded that the animals receiving YHY experienced a higher-than-expected oral bioavailability of mycotoxins, due to an indirect effect of YHY on intestinal health of the piglets. Evaluation of two mycotoxin detoxifiers that contained yeast cells or esterified glucomannans derived from yeast cell wall *S. cerevisiae* showed a significantly higher oral bioavailability for DON in broilers for the detoxifying groups compared to the negative control [37]. Furthermore, it was found that a glucomannan mycotoxin binder, in combination with T2-toxin, enhanced the oral absorption of the antibiotic doxycycline in pigs [38]. This observation was confirmed in another study that showed a significant influence of a mycotoxin detoxifying agent, consisting of bentonite-montmorillonite clay with a yeast, on the oral absorption of oxytetracycline in broiler chickens [39]. Although the mechanism is still unclear, it was hypothesized that the mycotoxin binder indirectly promoted intestinal health, changed intestinal immunological parameters, or influenced intestinal mucus production, albeit after a short-term exposure of the animal to the yeast [37]. A recent investigation using the probiotic yeast *S. cerevisiae* var. *boulardii* reported that the dietary administration of the probiotic to piglets fed a diet contaminated with DON at 3 mg/kg resulted in significantly lower histological alteration in the intestine, suggesting a better intestinal health, while the effect of DON on plasma metabolome and histological alterations in liver and kidney were attenuated by the yeast. Even if the modes of action involved between a probiotic and mycotoxin binders are likely to be different it appears difficult to conclude that an improved intestinal health would result in an increase oral bioavailability of mycotoxins [40].

Although within the context of this experiment and the described limitations, the YHY prototype was not an effective agent for mycotoxin adsorption, it is important to note that the piglets were administered with a multiple mycotoxin bolus that resembled a feed contamination amount of 1 mg/kg for DON and OTA, and 10 mg/kg for ZEA. This dose was realistic for DON under conventional conditions, whereas for OTA and ZEA, it greatly exceeded (a factor 20 and 100 above EU guidelines, respectively) mycotoxin levels commonly found in feed and raw materials of different geographical origin [4]. Interestingly, our results showed that YHY reduced the absorption of DON in pigs despite the presence of high concentrations of OTA and ZEA, also suggesting that YHY exerted different binding mechanisms for these mycotoxins. The observation of simultaneous binding of mycotoxins by YHY is of particular importance, since animals are generally exposed to multiple mycotoxins present in the feed under field conditions [41].

Additionally, in our study, the influence of feed matrix was not taken into account. Interestingly, it has been suggested that pre-incubation of binder and mycotoxin (as in feed under field conditions) could lead to a more efficient adsorption for B-glucan: i.e., pre-incubation led to a rapid and better reduction of DON absorption in vitro using a Caco-2 cells model [22].

To the best of the authors’ knowledge this is the first study specifically addressing the efficacy of a single yeast hydrolysate ingredient in a toxicokinetic model on the capacity to adsorb multiple mycotoxins in piglets. The outcomes of our toxicokinetic study indicate the need to further improve the experimental model and for a more thorough in vivo evaluation of the efficacy of YHY using natural mycotoxin contaminated feed to evaluate its capability in reducing mycotoxicoses in farm animals under field conditions.

## 4. Materials and Methods

### 4.1. Preparation of the Yeast Products

The *S. cerevisiae* strains tested in vitro were selected based on distinct physiological properties related to yeast cell wall and sterol metabolism (proprietary information). The enzymatic yeast hydrolysate (designated YHY) was prepared from cream of *S. cerevisiae* strain CNCM I-5405 that was heated to 50 °C, and papain (Promod^TM^, Biocatalysts, Cardiff, UK) was added. The mixture was incubated for 15 to 20 h at a pH of above 5 for hydrolysis. Next, the mixture was heated for 1 h at a temperature above 70 °C to inactivate all enzyme activity. The pH of the mixture or hydrolysate was then adjusted with NaOH to 6.0, heated to 75 °C for 60 s, and dried by roller drying into powder. The yeast autolysates YA1 and YA2 were prepared from cream of *S. cerevisiae* strains LYCC6988 and LYCC6382, respectively, without recourse to exogeneous enzymes, and dried by roller drying into powder.

### 4.2. In Vitro Assessment of pH-Dependent Adsorption/Desorption

All chemicals and solvents used were purchased from Sigma-Aldrich (St. Louis, MO, USA) and were of analytical grade. The adsorption test was performed essentially as described in [21] with DON, ZEA, and OTA concentration in the test buffer of 1, 0.5, and 0.1 ppm. The yeast products were resuspended in PBS solution (CaCl_2_·2H_2_O, 1.2 mM; KCl, 2.7 mM; KH_2_PO_4_, 1.5 mM; MgCl_2_·6H_2_O, 1.1 mM; NaCl, 138 mM; Na_2_HPO_4_·2H_2_O, 8.1 mM; pH = 5.0) to reach a final concentration 2 mg/mL. The pH of the mixtures was adjusted to 3.0 with 1 M HCl and incubated at 37 °C for 2 h under constant agitation to simulate the pH condition during gastric passage in a monogastric animal. After this first incubation step, a sample was taken for further analysis. The incubations were continued in the same flask by raising the pH to pH 5.0 with 1 M NaOH and leaving the incubation mixture for 2 h under constant agitation at 37 °C. After sampling, incubations were continued at a final pH of 8.5. These latter two incubation steps simulate the pH conditions during intestinal passage of a monogastric animal. As a negative control, the PBS solution with mycotoxin and without adsorbent was incorporated in the assay. Samples were immediately centrifuged to separate the binder from the aqueous phase and the supernatants were stored at −20 °C until further analysis by LC-MS/MS. All adsorption tests were carried out in triplicate. The amount of mycotoxin adsorbed by the yeast products was calculated according to the difference between the initial and final concentration of the mycotoxin in the supernatant. Data was analyzed in SAS Studio version 9.4 (SAS Inst. Inc., Cary, NC, USA) using a mixed model with pH and binder as fixed effect. To determine the differences between the binder and pH levels, the Tukey test was used. For all outcomes, statistical significance was declared where *p* ≤ 0.05.

### 4.3. Quantification of Mycotoxinsin Supernatants of Adsorption Tests

The mycotoxin concentration in the supernatants was determined by ultra-high performance liquid chromatography with mass spectrometry (LC-MS/MS) (ThermoFischer Scientific Inc., Waltham, MA, USA). For this purpose, 20 µl of an internal standard solution is added to 200 µl of supernatant, mixed by vortex (15 s), and transferred into an autosampler vial. As internal standards were used, ^13^C_15_–DON (25 µg/mL), ^13^C_20_–OTA (10 µg/mL), and ^13^C_18_ –ZEA (25 µg/mL) (Romer Laboratories, Tulln, Austria) in acetonitrile was used. An aliquot of 5 µl was analyzed using a TSQ^®^ Thermo Endura^TM^ LC-MS/MS equipped with a quaternary, low-pressure mixing Ultimate 3000 pump, a column oven (40 °C ± 1 °C), and a temperature controlled autosampler (ThermoFischer Scientific Inc., Waltham, MA, USA). Chromatographic separation of aliquots was achieved on a reversed phase C18 Ascentis Express column (75 mm × 2.1 mm internal diameter, particle diameter: 2.7 µm), and a guard column of the same type (Sigma-Aldrich, St. Louis, MO, USA). A gradient elution program was performed with 5% (*v*/*v*) acetic acid and 0.03854% (*w*/*v*) ammonium acetate in water and in methanol. Water, methanol, acetic acid and acetonitrile were of LC-MS grade (VWR International B.V, Amsterdam, The Netherlands). The LC column effluent was interfaced to a TSQ^®^ triple-stage quadrupole mass spectrometer equipped with a heated electronspray ionization (h-ESI) probe (ThermoFischer Scientific Inc., Waltham, MA, USA). The mass spectrometer was operated in the multiple reaction monitoring mode with two ion transitions for each mycotoxin for identification and quantification. LC-MS/MS instrument control and data processing was performed using ThermoScientific^TM^ Dionex^TM^ Chromeleon^TM^ Chromatography Data System Version 7.2.9 (Thermo Fischer Scientific Inc., Waltham, MA, USA). The limit of quantification (LOQ) of DON, ZEA, and OTA was 5, 5, and 1 ng/mL, respectively, whereas the limit of detection (LOD) was 2, 2, and 0.5 ng/mL, respectively.

### 4.4. Animal Study

Twelve female piglets (breed: Hybrid), of 15–20 kg bodyweight (BW), were randomized on arrival in such a way that two groups, each with six animals, were formed with about the same average BW/group. The administration of mycotoxins and the binder YHY to the piglets was as described in [23]. The piglets received commercial pig feed during the acclimatization period (7 days). This feed was also given during the study, except for 12 h prior to bolus administration to 4 h post administration. The feed was given ad libitum. The commercial feed was evaluated on contamination with mycotoxins by a multi-mycotoxin LC-MS/MS method at Primoris (Zwijnaarde, Belgium).

The piglets were administrated a single oral bolus of 0.05 mg DON/kg BW, 0.05 mg OTA/kg BW, and 0.5 mg ZEA/kg BW, by oral gavage using an intragastric tube. This dose resembles a feed contamination amount of 1 mg/kg for DON and OTA, and 10 mg/kg for ZEA. The EU recommendation for maximal tolerable concentration in pig feed for DON, OTA, and ZEA is 0.9, 0.05, and 0.1 mg/kg, respectively [18]. All mycotoxins used in the animal study were purchased from Fermentek^®^ (Jerusalem, Israel). DON (1 mg/mL), OTA (1 mg/mL), and ZEA (10 mg/mL) were dissolved in ethanol and further diluted with tap water to a volume of 10 mL. Six of the twelve piglets received the mycotoxin bolus in combination with the yeast hydrolysate (100 mg/kg BW), resembling an inclusion amount of 2 g/kg feed), suspended in 10 mL of tap water, in a parallel study design. Immediately after administration of the bolus, the intragastric tube was rinsed with 50 mL of tap water. Blood samples (2 mL), drawn from the *vena jugularis externa*, were taken in EDTA tubes and centrifuged (2851× *g*, 10 min, 4 °C) to obtain plasma. The timepoints of blood sampling were 0 h (just before administration) and 0.25, 0.5, 0.75, 1, 1.5, 2, 3, 4, 6, 8, 12, 24, 48, 72, and 96 h post administration (p.a.). Aliquots (250 µL) of plasma samples were stored at −20 °C until analysis.

### 4.5. Quantification of Mycotoxins in Plasma

The analysis of DON, OTA, ZEA and the main phase II metabolite of ZEA, namely ZEA-GlcA, in plasma was performed by using a validated UHPLC-MS/MS method [42]. For ZEA-GlcA peak areas were corrected for the internal standard (IS, ^13^C-ZEA) and results are presented as peak area ratio (=peak area ZEA-GlcA/peak area IS).

### 4.6. Toxicokinetic Analysis

Toxicokinetic modeling of the plasma concentration-time profiles of DON, OTA, and ZEA-GlcA was done by non-compartmental analysis (Phoenix 8.1, Pharsight Corporation, Sunnyvale, CA, USA). No ZEA was quantifiable in plasma, hence the use of ZEA-GlcA as biomarker for exposure. The following parameters were calculated: area under the curve from time zero to 0.5, 2, 4, 8, and/or 96 h (AUC_0–0_._5 h,_ AUC_0–2 h_, AUC_0–4 h,_ AUC_0–8 h_ or AUC_0–96 h_); maximal plasma concentration (DON and OTA) or maximal plasma chromatographic peak area (ZEA-GlcA) (C_max_); time at C_max_ (T_max_), elimination half-life time (T_1/2el_); and elimination rate constant (k_el_).

### 4.7. Effect of the Mycotoxin Binder on Oral Absorption of the Mycotoxins

The relative oral bioavailability expressed as a percentage, F = ([AUC_0–0.5/2/4/8 h or 96 h_ mycotoxin + binder/AUC_0–0.5/2/4/8 h or 96 h_ mycotoxin]×100), was evaluated as marker for efficacy of the mycotoxin binder. The effect of the mycotoxin binder on the oral absorption of the mycotoxin was evaluated by comparing the above described toxicokinetic parameters between the mycotoxin and mycotoxin plus binder treated piglets. A one-way ANOVA was performed (SPSS, IBM, Armonk, NY, USA), with the LSD-test as post hoc test, to evaluate possible significant differences between groups for each toxicokinetic parameter. Homogeneity of variances was first evaluated using the Levene’s test. When variances were not homogeneous, data were log-transformed. The level of significance was set at *p* ≤ 0.05.

## Figures and Tables

**Figure 1 toxins-14-00007-f001:**
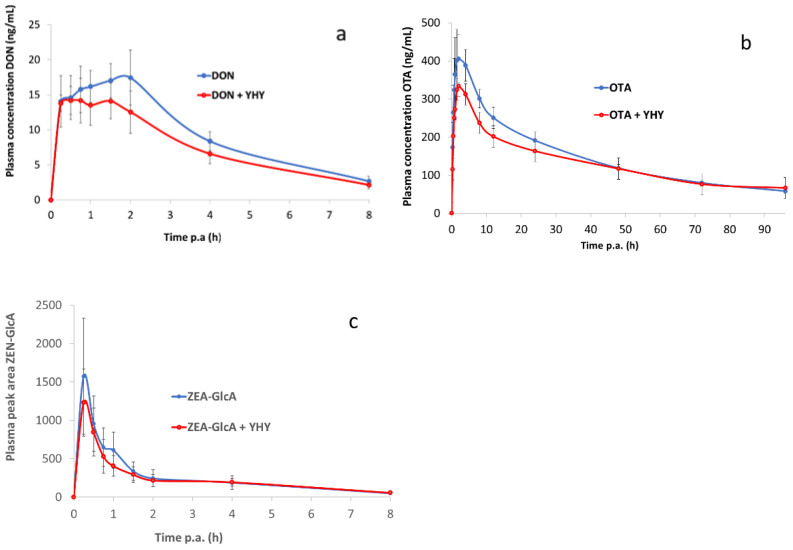
Plasma concentration-time profile of mycotoxins in piglets after single oral doses of DON (0.05 mg/kg BW) (**a**), OTA (0.05 mg/kg BW) (**b**), and ZEA (0.5 mg/kg BW) (**c**), alone or in combination with yeast hydrolysate YHY (0.1 g/kg BW). Values are presented as mean (*n* = 6) + SD. p.a: post administration.

**Table 1 toxins-14-00007-t001:** Concentration of DON, OTA, and ZEA in supernatants (ng/mL) and percent adsorption (%) of mycotoxins by three yeast product treatment groups in an in vitro model designed to mimic the gastric passage of a monogastric animal. The values are indicated as means ± standard deviation (SD) of three independent experiments *.

**DON**									
**pH 3.0**			**pH 5.0**			**pH 8.5**			
Treatment	ng/mL	% of NC	*p* Value	(ng/mL)	% of NC	*p* Value	ng/mL	% of NC	*p* Value
NC	990 ± 26	---	---	992 ± 30	---	---	1012 ± 70	---	---
YHY	540 ± 5	45.5 ± 1.0	<0.0001	538 ± 7	45.8 ± 0.8	<0.0001	543 ± 20	46.7 ± 1.8	<0.0001
YA1	887 ± 36	10.3 ± 1.7	0.3133	890 ± 5	10.0 ± 2.1	0.3345	903 ± 92	11.4 ± 2.1	0.2551
YA2	876 ± 45	11.4 ± 2.4	0.2049	878 ± 26	11.3 ± 1.9	0.2049	963 ± 90	4.8 ± 3.6	0.9802
**OTA**									
NC	32 ± 6	---	---	57 ± 8	---	---	68 ± 6	----	---
YHY	31 ± 3	9.3 ± 8.1	1.0000	52 ± 3	8.3 ± 7.6	0.9943	60 ± 5	12.0 ± 3.6	0.8283
YA1	25 ± 5	21.0 ± 7.6	0.9525	50 ± 9	14.0 ± 13.5	0.9525	66 ± 5	8.3 ± 7.6	0.9998
YA2	27 ± 8	16.3 ± 14.8	0.9943	52 ± 6	8.3 ± 7.6	0.9943	65 ± 5	5.0 ± 4.4	0.9998
**ZEA**									
NC	275 ± 6	---	---	280 ± 5	---	---	376 ± 38	---	---
YHY	264 ± 13	4.0 ± 3.5	0.9998	247 ± 11	11.7 ± 2.0	0.6119	328 ± 45	13.0 ± 2.7	0.1427
YA1	125 ± 9	54.3 ± 4.0	<0.0001	120 ± 10	57.0 ± 3.1	<0.0001	188 ± 8	49.7 ± 5.5	<0.0001
YA2	109 ± 5	60.3 ± 0.6	<0.0001	107 ± 16	62.0 ± 6.2	<0.0001	133 ± 8	64.7 ± 3.2	<0.0001

* Calculated in comparison to control incubations without any adsorbent. *p* values were calculated with SAS Studio using the actual DON concentration value in assay solutions.

**Table 2 toxins-14-00007-t002:** Major toxicokinetic characteristics of DON after single oral bolus administration, with and without the binder YHY. Values are presented as mean (*n* = 6) ± SD.

Toxicokinetic Parameters	DON	DON + YHY	*p* Value
AUC_0–8 h_ (h.ng/mL)	74.57 ± 8.39	60.03 ± 12.58	0.462
AUC_0–2 h_ (h.ng/mL)	30.03 ± 3.87	25.76 ± 4.58	0.561
C_max_ (ng/mL)	20.17 ± 4.22	15.47 ± 1.83	0.306
T_max_ (h)	1.13 ± 0.71	0.88 ± 0.29	0.626
T_1/2 el_ (h)	2.15 ± 0.10	2.39 ± 0.25	0.479
k_e_ (1/h)	0.32 ± 0.01	0.30 ± 0.03	0.377
Relative F AUC_0–8 h_ (%)	/	80.50	
Relative F AUC_0–2 h_ (%)	/	85.79	

AUC_0-t:_ area under the plasma concentration-time curve from time 0 to 2 or 8 h post administration; C_max_, maximum plasma concentration; T_max_, time at maximum plasma concentration; T_1/2 el_, elimination half-life; k_e_, elimination rate constant; and Relative F, relative oral bioavailability.

**Table 3 toxins-14-00007-t003:** Major toxicokinetic characteristics of OTA after single oral bolus administration, with and without the binder YHY. Values are presented as mean (*n* = 6) ± SD.

Toxicokinetic Parameters	OTA	OTA + YHY	*p* Value
AUC_0–96 h_ (h.µg/mL)	14.13 ± 1.48	12.28 ± 2.06	0.135
AUC_0–4 h_ (h.µg/mL)	1.42 ± 0.24	1.14 ± 0.25	0.288
C_max_ (ng/mL)	426.23 ± 85.10	353.83 ± 64.21	0.076
T_max_ (h)	2.42 ± 1.06	2.04 ± 0.65	0.853
T_1/2 el_ (h)	45.12 ± 6.61	50.69 ± 12.40	0.464
k_e_ (1/h)	0.016 ± 0.002	0.015 ± 0.004	0.674
Relative F AUC_0–96 h_ (%)	/	86.86	
Relative F AUC_0–4 h_ (%)	/	79.86	

Abbreviations are as in Table 2.

**Table 4 toxins-14-00007-t004:** Major toxicokinetic characteristics of ZEA-GlcA after single oral bolus administration, with and without the binder YHY. Values are presented as mean (*n* = 6) ± SD.

Toxicokinetic Parameters	ZEA	ZEA + YHY	*p* Value
AUC_0–8 h_ (h.peak area/mL)	2168.20 ± 494.87	1917.16 ± 444.98	0.985
AUC_0–0_._5 h_ (h.peak area/mL)	505.92 ± 227.89	407.48 ± 141.70	0.765
C_max_ (peak area/mL)	1627.15 ± 737.12	1302.53 ± 485.21	0.308
T_max_ (h)	0.33 ± 0.14	0.29 ± 0.07	0.482
T_1/2 el_ (h)	3.60 ± 2.00	2.93 ± 0.85	0.523
k_e_ (1/h)	0.27 ± 0.12	0.27 ± 0.05	0.533
Relative F AUC_0–8 h_ (%)	/	88.42	
Relative F AUC_0–0_._5 h_ (%)	/	80.54	

Abbreviations are as in Table 2.

## Data Availability

Data are available upon request; please contact the contributing authors.

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
