# Peer review of "Evaluation of a Yeast Hydrolysate from a Novel Strain of Saccharomyces cerevisiae for Mycotoxin Mitigation using In Vitro and In Vivo Models"

_toxins, 2021, doi:10.3390/toxins14010007_

Round 1

Reviewer 1 Report

Within the reviewed manuscript entitled „Evaluation of a yeast hydrolysate from a novel strain of Saccharomyces cerevisiae for mycotoxin mitigation using in vitro and in vivo models” the authors report their work regarding a yeast hydrolysate and how it affects mycotoxin levels either in an in vitro experimental setup as well as within a toxicokinetic in vivo study.

The research presented here focuses on one out of three yeast hydrolysates and its effect on the mycotoxin levels DON, ZEA and OTA. The adsorptive effect of yeast hydroslysates regarding mycotoxins mitigating the toxin concentration in food/feed is not new and there are several commercial products on the marked that can be applied to feed to reduce mycotoxin uptake. However identifying new adsorptive and thus effective agents that could significantly reduce mycotoxin uptake by livestock would be beneficial. Unfortunately the study presented here does not describe an effective yeast hydrolysate. Already within the abstract the authors admit that their yeast hydrolysate (YHY) only caused a non-significant reduction in oral availability of the respective mycotoxins. Nevertheless they claim that YHY may have the potential to reduce the absorption of mycotoxins in animals. In my opinion this is not supported by the presented results. The authors themselves conclude the need to further improve the experimental model and for more thorough in vivo evaluation.

Hence, in my opinion the authors should focus on such experimental improvements especially with respect to statistically significance. I would therefore not recommend the manuscript for publication in its current form.

There are several major weaknesses that need to be addressed. First the authors investigated three types of yeast hydrolysates in vitro and their effect on DON. They identified YHY which leads to DON reduction of 45%. The authors then employed this YHY in their animal study and analyzed the effect on DON as well as OTA and ZEA within the animals. It is not clear why the authors did not show any experimental data regarding the effect of their YHY on OTA and ZEA in an in vitro approach. Only if such an in vitro experiment would have shown an effect on the respective mycotoxins further in vivo studies would have made sense. Instead the authors applied all three mycotoxins simultaneously in very high dosage to a group of six pigs and investigated how the mycotoxin levels decreased within several hours after gavage either with or without the YHY. This is questionable because it cannot be excluded that the mycotoxin levels and/or YHY affects each other somehow. There should have been toxicokinetic studies with all mycotoxins separately in different groups of animals.

Typically animal experiments are highly regulated and there are regulations ensuring certain ethical standards. The authors should mention under which circumstances and regulations they did their animal study and what health risks were associated with their study especially when feeding mycotoxin levels exceeded a factor of 20 and 100 above EU guidelines as the authors mention.

The authors further mention that they found out that the regular food of the animals applied was itself contaminated with mycotoxins. This should in all cases be avoided in a new study to exclude any accumulative effects that might lead to deviant results.

All experimental data of the in vivo study showed very high standard deviation that makes the observations (slight decrease in mycotoxin concentrations) statistically insignificant. Standard deviations shown within figure 1 are misleading since only the error bar over the curve is presented. The standard deviation should be indicated as +/- error bars.

All experiments should be supplemented with respective controls or the control experiment data should be added. E.g. in the tables data from negative controls (no mycotoxin) and maybe even positive controls (e.g. commercial mycotoxin adsorbing agents as mentioned in Reference [29]) would have improved experimental data.

Reviewer 2 Report

Here is an explanation of my assessment:

  • Abstract is confusing.
    • Some abreviations appear before they are explained;
    • no quantitative result and / or conclusion has been singled out, and this is not a qualitative study.
    • Same goes for conclusion as well;
      • with two sentences in five lines only points out that "To the best of the authors' knowledge, this is one of the few studies, and the first study to specifically evaluate a single yeast hydrolyzate ingredient, in which a toxicokinetic study was used to test its ability to adsorb more mycotoxins in piglets." .... and that's it ?!
      • If the study is unique, then it should certainly be more clearly described and pointed out specifically what has been observed with a decrease in systemic exposure to DON, OTA and ZEA and what happens with the addition of YHY in the model of pigs with oral bolus.
  • If this is an original scientific study and not a scientific line, then this is how it should be approached.
  • The results are valuable, but throwing in the results, retelling them - is not a scientifically acceptable form.
  • Updated references are used (see no. 9; 11; 12; 27; all from the same group of authors: Jouany, J.-P.; Yiannikouris, A.; Bertin, G)
  • For some references is not clear when they are published (No. 16)
  • Conclusion?!
  • Written text in lines 270-275 is not a suitable text given as conclusion of a professional paper!

This is the reason why I rated this article "needed major revision".

Reviewer 3 Report

The authors evaluated the Evaluation of a yeast hydrolysate from a novel strain of Saccharomyces cerevisiae for mycotoxin mitigation using in vitro and in vivo models.

The paper is well written and well organized.

Some minor remarks are the folowing.

Line 31. You can also add the very recent publication entitled Agriopoulou, S.; Stamatelopoulou, E.; Varzakas, T. Advances in Occurrence, Importance, and Mycotoxin Control Strategies: Prevention and Detoxification in Foods. Foods 20209, 137. https://doi.org/10.3390/foods9020137

Line 164. in vitro and in vivo...Please write in italics.

Line 179. in vitro...Please write in italics.

Line 186.in vitro...Please write in italics.

Line 190. S. pastorianus....Please write in italics.

Line 231.S. cerevisiae....Please write in italics.

Please write again the conclusions with the main findings of the study.

Reviewer 4 Report

In this study, the authors evaluated the efficiency of three selected yeast products as mycotoxin binder using in vitro and in vivo models. The paper is well organized, has clear objectives and the drawn conclusions are coherent with the obtained results. The results of this work indicate that YHY may have the potential to reduce the absorption of multiple mycotoxins in animals after exposure.

Line 19: To arrange alphabetically the key-words

Introduction: Well written!

Results: Well written!

Lines 174 – 176: I would like to suggest to modify these sentences in this way in order to widen the discussion…”Yeasts and yeast cell wall extracts have shown in vitro adsorption efficacy for a number of mycotoxins, including ZEA, AFB1, T2-toxin, patulin and OTA [8,11-13,28] as well as in fungi (Bosso et al., 2016;

Bosso, L., Scelza, R., Varlese, R., Meca, G., Testa, A., Rao, M.A., Cristinzio, G. 2016. Assessing the effectiveness of Byssochlamys nivea and Scopulariopsis brumptii in pentachlorophenol removal and biological control of two Phytophthora species. Fungal Biology, 120, 645-653.

Wang, X., Bai, Y., Huang, H., Tu, T., Wang, Y., Wang, Y., ... & Su, X. (2019). Degradation of aflatoxin B1 and zearalenone by bacterial and fungal laccases in presence of structurally defined chemicals and complex natural mediators. Toxins, 11(10), 609.

Lines 270 – 275: Please, expand you conclusions

Lines 303 – 304: I think that you should add this recent references as example to support your methods “Samples were immediately centrifuged to separate the binder from the aqueous phase and the supernatants were stored at -20 °C until further analysis by LC-MS/MS.”. I would like to suggest:

Bosso, L., Lacatena, F., Cristinzio, G., Cea, M., Diez, M.C., Rubilar, O. 2015. Biosorption of pentachlorophenol by Anthracophyllum discolor in the form of live fungal pellets. New Biotechnology, 32, 21-25.

Pradeep, S., Josh, M. S., Balachandran, S., Devi, R. S., Sadasivam, R., Thirugnanam, P. E., ... & Benjamin, S. (2014). Achromobacter denitrificans SP1 produces pharmaceutically active 25C prodigiosin upon utilizing hazardous di (2-ethylhexyl) phthalate. Bioresource technology, 171, 482-486.

Round 2

Reviewer 1 Report

The authors have added some of the required information/additional data/controls. They further added information regarding animal treatment with mycotoxins. However this does not change my opinion that the whole study presents statistically not significant data making their conclusion that YHY may have the potential to reduce the absorption of multiple mycotoxins in animals not convincing. There are still several methodological weaknesses in the study as feeding the animals with mycotoxin contaminated food in advance of the study and feeding mixtures of all three mycotoxins instead of feeding each toxin separately. Hence, in my opinion this study is still not sufficient for a publication in Toxins. However since the other reviewers seem to have a different opinion it is up to the editor to decide whether this topic and the study with its weaknesses in the present form is worth publishing. If the manuscript will be published the authors still need to take a closer look to the diagrams and the +- standard deviation. The error bars still look strange not presenting equal deviation above and below the curve (why are the error bars in some cases asymmetric?). For better visibility and to distinguish them the error bars should be colored according to the respective curve either in red or blue.

Reviewer 2 Report

Changes made by the authors have increased the quality of the paper.

Round 3

Reviewer 1 Report

no further comments to the authors